# Strength-Dependent Differences in the Magnitude and Time Course of Post-Activation Performance Enhancement in High Jump Athletes

**DOI:** 10.3390/jfmk10030333

**Published:** 2025-08-29

**Authors:** Javier Sanchez-Sanchez, Alejandro Rodríguez-Fernández

**Affiliations:** 1Research Group Planning and Assessment of Training and Athletic Performance, Universidad Pontificia de Salamanca, 37007 Salamanca, Spain; 2VALFIS Research Group, Institute of Biomedicine (IBIOMED), Faculty of Physical Activity and Sports Sciences, Universidad de León, 24071 León, Spain

**Keywords:** resistance training, muscle strength, athletic performance, acute performance response

## Abstract

**Background**: A post-activation performance enhancement (PAPE) can acutely improve explosive actions, but its time course may be influenced by individual strength levels. **Objectives**: The aim of this study was to analyze the performance responses following three PAPE protocols, considering the strength level as a modulating factor in trained high jump athletes. **Methods**: Twenty-one male high jumpers (Tier 3) were divided into stronger (SG, n = 10) and weaker (WG, n = 11) groups based on the median load (80 kg) lifted at 0.8 m/s in a velocity-based half-squat test. The participants completed three squat-based PAPE protocols (velocity loss thresholds of 5%, 10%, and 15%) in a randomized, double-blind crossover design. Their performance in a 10 m sprint (S10) and a countermovement jump (CMJ) was assessed at baseline and 0, 4, 8, and 12 min post-intervention. **Results**: No significant three-way interactions were observed for the S10 or CMJ performance (*p* > 0.05). The absolute CMJ performance was consistently higher in the SG across all the time points (*p* < 0.001, d = 1.25, large), with significant peak values observed at 4 min post-activation. However, both groups exhibited transient improvements in their S10 and CMJ performance that were statistically significant (*p* < 0.05) and of a large magnitude (d = 1.93–3.15), observed at 4 and/or 8 min post-activation, which subsequently declined by 12 min. **Conclusions**: The strength level modulates both the time course and the magnitude of the PAPE. Stronger athletes responded better to both less and more demanding protocols (5% to 15% velocity loss thresholds) with a 4–8 min recovery, whereas weaker athletes benefited mainly from less demanding stimuli (5% velocity loss thresholds), provided that the recovery was sufficient (≈4 min) to allow potentiation to emerge. However, with more demanding protocols (15% velocity loss thresholds), a longer recovery period (≈8 min) appears necessary.

## 1. Introduction

Within the field of high-performance sports, it is increasingly acknowledged that competitive outcomes are not determined solely by the volume and quality of physical and technical training [1]. Instead, a wide range of complementary factors play a crucial role in an athlete’s ability to adapt and perform under demanding conditions [2]. Components such as adequate recovery, optimal sleep, balanced nutrition, psychological readiness, and stress regulation are now considered essential to fully assimilate training stimuli and sustain performance [1]. In addition to these factors, more immediate and practical strategies—such as brief pre-activation protocols designed to acutely enhance neuromuscular function—have gained relevance due to their potential to improve performance, particularly in explosive and high-intensity competitive actions [1,3]. Within this context, acute neuromuscular enhancement methods have become increasingly prominent, with post-activation potentiation (PAP) standing out as a well-documented approach aimed at eliciting short-term, but meaningful, improvements in performance outcomes through the application of prior high-intensity conditioning activity [4].

Although the term PAP has traditionally been used to describe acute improvements in muscular performance following a prior contraction, several authors have highlighted its limited applicability in real competitive settings, as it is primarily associated with protocols involving brief, electrically induced twitch contractions [5]. In contrast, in applied training contexts where voluntary and functional movements predominate, the term post-activation performance enhancement (PAPE) has been proposed to more accurately define the immediate performance gains observed after a voluntary conditioning activity [6]. This effect is believed to result from several physiological mechanisms that are not yet fully understood, including the phosphorylation of myosin regulatory light chains—which increases calcium sensitivity and enhances force production [7]—as well as acute changes in muscle temperature, intracellular water content, and neuromuscular activation [8]. These internal processes give rise to an acute performance enhancement, the magnitude and duration of which are influenced by multiple variables, among which the individual characteristics emerge as primary modulators of the overall response [9].

Among these individual characteristics, factors such as the training status, the muscle composition—particularly the fiber type distribution—and the strength levels play a decisive role in modulating the balance between fatigue and potentiation, thereby influencing both the magnitude and timing of the PAPE response [10]. This marked inter-individual variability presents a significant challenge for standardizing optimal protocols across athletes with different profiles. Consequently, it has drawn considerable attention from researchers and practitioners aiming to clarify the role of the strength level, which has consistently emerged as a key modulator of PAPEs [11,12]. For example, a significant negative correlation (*r* = −0.771, *p* < 0.01) has been reported between the relative strength, assessed as the one-repetition maximum (1RM) in the back squat divided by the body mass, and the optimal recovery duration in recreationally trained individuals, along with a 7.1% increase in the peak power, regardless of the rest period [13]. In rugby athletes, stronger players, defined as those with a relative 1RM back squat ≥2× body mass, have shown significant potentiation from 3 to 12 min post-activation (i.e., peak at ~6 min), whereas weaker players, with a relative 1RM back squat <2× body mass, tend to peak later (i.e., ~9 min), with the overall effects being greater in the stronger group (*p* < 0.05) [14]. Similarly, meaningful improvements in the countermovement jump (CMJ) height (d > 0.9, *p* < 0.001) and 10 m sprint performance (d > 0.5, *p* < 0.05) have been observed following both flywheel and traditional activation strategies, particularly when the relative strength exceeds 2.0× the body mass [15]. In elite sprinters, their performance improved across the strength strata after heavy back squats (*p* < 0.05), with peak responses occurring at ~6 min in stronger athletes (≥2.5× body mass) and ~3 min in weaker ones, and with the stronger group exhibiting greater overall effects [12]. In contrast, some studies have found no consistent predictive relationship between the maximal dynamic strength or training background and PAPE expression, highlighting substantial individual variability and the need to identify responders on a case-by-case basis [16].

Taken together, the available evidence highlights the complexity of the PAPE phenomenon and the substantial inter-individual variability in both its magnitude and timing. This variability hinders the development of standardized activation protocols and reinforces the importance of tailoring the load, volume, and recovery to each athlete’s strength profile. To optimize the performance outcomes in sport-specific contexts, it is therefore essential to deepen our understanding of how individual factors, particularly the strength level, influence the PAPE response. Therefore, the aim of this study was to analyze the performance responses following three PAPE protocols, considering the strength level as a modulating factor in trained high jump athletes.

## 2. Materials and Methods

### 2.1. Participants

A total of 21 male high jump athletes participated in this study. All were members of the same athletics club and, according to the classification framework proposed by McKay et al. [17], were categorized as Tier 3 (trained/developmental athletes). Based on the load lifted at a bar speed of 0.8 m/s in a velocity-based squat test, the participants were divided into a stronger group (SG, n = 10; lifted load: 96.5 ± 7.09 kg; age: 20.9 ± 0.4 years; height: 183.6 ± 6.2 cm; body mass: 71.8 ± 6.1 kg; personal best: 1.93 ± 0.08 m) and a weaker group (WG, n = 11; lifted load: 74.1 ± 5.39 kg; age: 20.7 ± 0.5 years; height: 179.4 ± 5.7 cm; body mass: 66.7 ± 7.7 kg; personal best: 1.83 ± 0.10 m). A preliminary statistical analysis revealed no significant differences between the participants in terms of their age, height, body mass, or personal best (*p* > 0.05). However, significant differences (*p* < 0.01) were found in their strength levels. The inclusion criteria required participants to belong to the same athletics club and to have accumulated at least 5 years of experience in athletics, completed a minimum of 3 years of structured resistance training, and remained injury-free in the 3 months prior to the study. All the athletes participated regularly in 4 to 6 weekly training sessions that combined technical drills; plyometric exercises; multi-joint strength training such as squats, deadlifts, and Olympic lifts; sprint work; mobility routines; and injury prevention exercises. The training sessions lasted between 45 and 90 min depending on the phase of the season. This study was conducted while following the Declaration of Helsinki guidelines, with all the athletes providing written informed consent after completing a health questionnaire.

### 2.2. Procedures

A randomized, double-blind, repeated-measures crossover design was employed to examine the effects of three different PAPE protocols, consisting of half-squat sets performed until a loss of 5% (PAPE5), 10% (PAPE10), or 15% (PAPE15) in movement velocity was reached. The load used for each protocol was individualized based on each participant’s load–velocity profile, corresponding to a mean propulsive velocity of 0.8 m/s. Each set was continued until the designated velocity loss threshold was attained. Prior to the experimental phase, all the participants completed a familiarization session. This session aimed to ensure the proper execution of the testing procedures, establish individual load–velocity profiles via progressive testing, and minimize potential learning effects. The half-squat technique was standardized using a knee angle of 90°, measured with a goniometer. Additionally, the participants were guided through the standardized warm-up and the testing protocols for the 10 m sprint (S10) and CMJ, ensuring a consistent technique across sessions. Based on the load corresponding to a mean propulsive velocity of 0.8 m/s, the participants were divided into a stronger group (SG) and a weaker group (WG), using the sample median of 80 kg as the cut-off point. The participants were blinded to their group allocation.

Each experimental session began with the same standardized warm-up, followed by a baseline performance assessment, which served as the control condition (CON), and then the assigned activation protocol. To preserve the double-blind design, neither the participants nor the performance assessors were aware of the specific velocity loss threshold applied in each session. Each PAPE protocol was implemented twice per participant—once followed by S10 testing and once by CMJ testing—resulting in a total of six testing sessions per individual. The order of conditions was counterbalanced across the participants, and a minimum of 72 h separated each session to ensure full recovery and minimize residual fatigue.

The performance outcomes (S10 and CMJ tests) were assessed at four time points following each activation protocol, which were immediately after the intervention (R0) and 4 min (R4), 8 min (R8), and 12 min (R12) later (Figure 1). All the testing sessions were conducted at the same time of day for each participant, under controlled environmental conditions, and with standardized verbal encouragement to ensure methodological consistency.

Finally, the percentage changes relative to the CON were calculated for each recovery interval, adapting the methodological approach proposed by Mola et al. [18]. The percentage change at each recovery point was calculated using the formula %ΔRx = (Post/Baseline) × 100. For S10, where a decrease in time reflects improvement, the formula was inverted and expressed as %ΔRx = (Time CON/Time Rx) × 100. For the CMJ height, the calculation followed the standard structure and was expressed as %ΔRx = (Height Rx/Height CON) × 100. In both cases, a value of 100% represents no change; values above 100% indicate a performance enhancement (i.e., potentiation); and values below 100% reflect a performance decrement or post-activation performance depression. This computation enabled a standardized comparison of the magnitude and time course of the performance response across recovery intervals, strength groups, and PAPE protocols.

### 2.3. Measurements

To determine the individualized load associated with a mean propulsive velocity of 0.8 m/s, the participants completed a progressive half-squat test on a Smith machine (BH L350B, BHFitness^®^, Vitoria-Gasteiz, Spain) with the bar restricted to vertical movement, following previously established protocols [19]. Before the test, all the athletes performed a standardized warm-up supervised by the same investigator, which included joint mobility exercises for both the upper and lower limbs and 2 × 8 rep with a 20 kg load, with 3 min of rest between sets [20]. During testing, the barbell was placed behind the neck at shoulder level with the feet shoulder-width apart, beginning and ending the movement with the hips and knees fully extended. The eccentric phase was performed under control until reaching a 90° knee flexion, previously determined with a goniometer and monitored using photoelectric cells that emitted an audible signal when the beam was interrupted. The descent velocity was maintained between 0.5 and 0.65 m/s [20], followed by a concentric phase performed with maximal voluntary intent until the full extension of the lower limbs was achieved. The initial set used a 20 kg load, allowing bar speeds above 1 m/s, and the load was increased in 10 kg increments thereafter. Each set included 3 continuous reps, with 4 min of rest between sets. The test concluded when athletes failed to reach a mean propulsive velocity ≥0.8 m/s across all 3 reps of a set, which corresponds to approximately 60% 1RM [21]. A linear encoder (SmartCoach Power Encoder^®^, SmartCoach Europe AB, Stockholm, Sweden) linked to the SmartCoach^®^ software (v3.1.3.09) was used to record the bar velocity and provide real-time feedback. The device sampled the vertical instantaneous velocity at 100 Hz using a 10-point rolling average filter and has previously shown a mean error of 0.52% ± 0.17% [22].

The sprint performance was assessed using an S10 test. The athletes initiated the sprint from a standing split stance, positioning their lead foot about 0.5 m behind the start line, and began at their own discretion. A dual-beam photocell system (Polifermo Light Radio, Microgate^®^, Bolzano, Italy) placed 90 cm above the ground (i.e., hip height) measured the sprint time. This system has a standard error of measurement of 0.03 s and a coefficient of variation (CV) of ~2% [23]. Due to study constraints, only one trial was performed per test session. The test showed a high reliability across 3 familiarization trials, with an intraclass correlation coefficient (ICC) of 0.97 and a mean CV of 0.45%.

The vertical jump performance was measured using the CMJ test. Athletes performed the jump on an optical contact mat system (OptoGait v.1.12.17.0, Microgate^®^, Bolzano, Italy) by following the protocol described in previous studies [24]. Arm swing was permitted, and the athletes were instructed to maintain a stable landing position after the jump. The jump height in cm was used as the primary outcome. Each athlete completed only one attempt due to design limitations [25]. The CMJ test demonstrated an excellent reliability across the familiarization trials, with an ICC of 0.98 [26].

### 2.4. Post-Activation Performance Enhancement Protocols

The CON, which did not involve any PAPE stimuli, was preceded by a general warm-up consisting of 5 min of cycling on a stationary bike at an intensity corresponding to 60–70% of the participant’s predicted maximum heart rate, which had been determined from previous field tests (multistage shuttle run test) routinely performed by the athletes as part of their physical condition monitoring. The heart rate was continuously monitored using a Polar Team Pro 2 sensor (Polar Electro^®^, Kempele, Finland), with real-time feedback displayed on a screen visible to the participant to help maintain the prescribed intensity.

For the experimental protocols, the participants performed half-squat exercises on a BH L350B Smith Machine (BHFitness^®^, Vitoria-Gasteiz, Spain), where the bar path was restricted to vertical movement. The load was individualized to correspond with an average propulsive velocity of 0.8 m/s, as determined previously in baseline testing. A linear position sensor (SmartCoach Power Encoder^®^, SmartCoach Europe AB, Stockholm, Sweden) paired with the SmartCoach™ software (v5.6.0.8) tracked the movement velocity continuously, allowing instant feedback.

To maintain a double-blind setup, one researcher programmed the target velocity loss threshold for the session (i.e., set at either 5%, 10%, or 15%) into the software before testing began. A second researcher, responsible for overseeing the exercise, was kept unaware of the programmed threshold and focused solely on monitoring technique and motivating the participants to perform each concentric phase with maximal effort. The software automatically terminated the set when the preselected velocity drop was detected, thereby eliminating the need for subjective decisions about when to stop. Neither the athletes nor the supervising researcher knew which velocity loss threshold was assigned during each session, reducing any potential bias from the expectations. The number of repetitions completed before reaching the velocity loss criterion was logged for each protocol condition.

### 2.5. Statistical Analyses

Descriptive data are expressed as the mean ± standard deviation (SD). After confirming normality using the Shapiro–Wilk test, parametric analyses were performed. To provide a comprehensive understanding of both absolute performance changes and relative adaptations to the activation protocols, two separate three-way mixed-design ANOVAs were conducted. To examine the effects on the S10 and CMJ performance, a three-way mixed-design ANOVA was conducted, including two within-subject factors—the condition (PAPE5, PAPE10, PAPE15) and the recovery interval (R0, R4, R8, R12)—and one between-subject factor—the group (SG vs. WG). To further explore the magnitude and temporal patterns of PAPE effects, additional three-way mixed-design ANOVAs were performed on the percentage change variables (%Δ) in the S10 and CMJ relative to baseline, using the same within- and between-subject factors: the condition (PAPE5, PAPE10, PAPE15), recovery interval (R0, R4, R8, R12), and group (SG vs. WG). The baseline values obtained after the standardized warm-up and prior to any activation protocol (control condition, CON) were used as reference values for descriptive comparison only. Mauchly’s test was used to verify the sphericity assumption. When this assumption was violated, the Greenhouse–Geisser correction was applied to adjust the degrees of freedom. In cases of significant interactions, Bonferroni-adjusted post hoc comparisons were performed. The effect sizes for the ANOVA were reported using partial eta squared (η_p_^2^), interpreted as small (<0.06), moderate (0.06–0.13), or large (≥0.14) [27], while Cohen’s d was used to quantify the magnitude of pairwise differences [28], classified as *trivial* (≤0.20), *small* (0.21–0.50), *moderate* (0.51–0.80), or *large* (>0.80) [27]. The statistical analyses were performed using SPSS, version 25.0 (IBM Corp., Armonk, NY, USA), with significance established at *p* < 0.05.

## 3. Results

In relation to the number of repetitions performed in each PAPE protocol, the SG completed 5.70 ± 1.42, 9.10 ± 1.97, and 12.2 ± 2.30 repetitions in PAPE5, PAPE10, and PAPE15, respectively, whereas the WG performed 6.18 ± 1.83, 10.3 ± 2.33, and 12.6 ± 2.77 repetitions.

Figure 2 presents the S10 performance results following each intervention. The repeated-measures ANOVA revealed no significant three-way interactions among group, PAPE protocol, and recovery time (F = 0.123, *p* = 0.998, η_p_^2^ = 0.003). Additionally, there were no significant main effects for the group (F = 0.1145, *p* = 0.892, η_p_^2^ = 0.001) or recovery time (F = 0.0358, *p* = 0.998, η_p_^2^ = 0.001). No performance differences were observed between the SG and WG at any post-activation time points. Furthermore, the baseline sprint performance under the CON condition did not differ across protocols or between strength groups.

Figure 3 presents the CMJ performance results following each intervention. The repeated-measures ANOVA revealed no significant three-way interactions among group, PAPE protocol, and recovery time (F < 0.001, *p* = 1.000, η_p_^2^ < 0.001). No significant main effects were found for the group (F < 0.001, *p* = 0.999, η_p_^2^ < 0.001) or time (F = 2.408, *p* = 0.050, η_p_^2^ = 0.033). Despite the lack of significant interactions, pairwise comparisons showed that the SG exhibited a significantly greater CMJ height than the WG at CON and R12 (*p* < 0.05), as well as at R4 and R8 (*p* < 0.01) across all protocols. Within-group comparisons revealed that, in the SG, the CMJ height at R4 was significantly higher than at R0 (*p* < 0.05, d = 2.02–2.05, *large*). No such differences were observed within the WG across the recovery time points.

Figure 4 presents the percentage changes in the S10 performance relative to the control condition following each intervention. The repeated-measures ANOVA revealed no significant three-way interactions among group, PAPE protocol, and recovery time (F = 0.364, *p* = 0.901, η_p_^2^ = 0.009). No significant main effects were found for the group (F = 0.2926, *p* = 0.831, η_p_^2^ = 0.004), and although the main effect of time approached significance (F = 4.3349, *p* = 0.014, η_p_^2^ = 0.037), post hoc comparisons revealed consistent patterns of performance enhancement. Under the PAPE5 condition, %ΔR4 and %ΔR8 were significantly greater than %ΔR0 in both the SG and WG (*p* < 0.01; d = 1.93–2.91, *large*), while %ΔR12 was significantly lower than %ΔR4 in both groups (*p* < 0.01, d = 2.28–2.29, *large*). Under the PAPE10 condition, %ΔR4 was significantly greater than %ΔR0 in the SG (*p* < 0.01, d = 2.48, *large*) and WG (*p* < 0.05, d = 2.51, *large*), and also greater than %ΔR12 in both groups (*p* < 0.01, d = 1.18–2.11, *large*). Under the PAPE15 condition, %ΔR4 exceeded %ΔR0 only in the WG (*p* < 0.01, d = 2.07, *large*), whereas %ΔR8 was significantly greater than %ΔR0 in both groups (*p* < 0.01; d = 2.27–3.15, *large*), and %ΔR12 was significantly lower than %ΔR8 in the WG (*p* < 0.01, d = 1.84, *large*).

Figure 5 presents the percentage change in the CMJ performance across the different PAPE protocols. The repeated-measures ANOVA revealed no significant three-way interactions among group, PAPE protocol, and recovery time (F = 0.006, *p* = 1.000, η_p_^2^ < 0.001). No significant main effects were observed for the group (F = 0.0206, *p* = 0.980, η_p_^2^ < 0.001), whereas the main effect of time was statistically significant (F = 8.872, *p* < 0.001, η_p_^2^ = 0.105). Under the PAPE5 condition, %ΔR4 and %ΔR8 were significantly greater than %ΔR0 in both the SG and WG (*p* < 0.01; d = 1.93–2.91, *large*), while %ΔR12 was significantly lower than %ΔR4 in both groups (*p* < 0.01; d = 2.28–2.29, *large*). Under the PAPE10 condition, %ΔR4 was significantly greater than %ΔR0 in the SG (*p* < 0.01, d = 2.51, *large*) and WG (*p* < 0.05, d = 2.48, *large*), and also greater than %ΔR12 in the SG (*p* < 0.01, d = 1.75, *large*) and WG (*p* < 0.01, d = 2.11, *large*). Under the PAPE15 condition, %ΔR4 exceeded %ΔR0 only in the WG (*p* < 0.01, d = 2.07, *large*), whereas %ΔR8 was significantly greater than %ΔR0 in both the SG and WG (*p* < 0.01; d = 2.27–3.15, *large*), and %ΔR12 was significantly lower than %ΔR8 only in the WG (*p* < 0.01, d = 1.84, *large*).

## 4. Discussion

The aim of this study was to analyze the performance responses following three PAPE protocols, considering the strength level as a modulating factor in trained high jump athletes. The results revealed that, while no significant differences in sprint performance (S10) were found between the stronger (SG) and weaker (WG) groups following any PAPE conditions, the SG consistently achieved greater CMJ heights than the WG across all the recovery time points (e.g., R4, R8, and R12) and protocols. In the SG, the absolute CMJ performance peaked at R4 post-activation with large effect sizes (d > 2.0). Regarding relative improvements (percentage change from baseline), both groups (SG and WG) exhibited significant improvements at R4 in both the S10 and CMJ following PAPE5 and PAPE10, and, in the case of the WG, also in PAPE15. At R8, the performance gains persisted in both the SG and WG for the PAPE5 and PAPE15 conditions. These findings suggest that squat-based PAPE protocols can effectively enhance explosive actions in high jumpers, regardless of an athlete’s strength level, although stronger athletes tend to exhibit superior baseline and post-activation CMJ values. The absence of significant group-by-time interactions for both the S10 and CMJ suggests that individual variability in the timing of the optimal recovery period may play a more critical role in maximizing the potentiation response than the strength status alone. From a practical standpoint, coaches should prioritize tailoring recovery intervals to the individual athlete rather than adjusting protocols solely based on maximal strength. This approach may be particularly valuable in short-term competitive settings or when integrating PAPEs into routine training plans to optimize acute performance.

Regarding the S10 performance, no significant effects were observed when analyzing the absolute sprint times, regardless of the group, protocol, or recovery time. However, when examining the percentage change from baseline (i.e., %ΔR), all the PAPE protocols elicited transient improvements at R4 and/or R8, followed by a performance decline at R12 in both PAPE5 and PAPE10. This decline was statistically significant, with %ΔR12 being lower than %ΔR4 in both groups (*p* < 0.01, d ≈ 2.2), confirming that potentiation was short-lived. In PAPE15, the WG showed significant improvements at R4 and R8 (*p* < 0.01), but these gains disappeared at R12. These findings support the notion that the conditioning activities can simultaneously trigger potentiation and fatigue, making recovery time a critical variable [12]. The improvements observed at R4 and R8 likely reflect a window during which potentiation exceeds fatigue, whereas the decline at R12 may indicate the dissipation of potentiation or the onset of delayed fatigue. Although some studies have reported that the effects of PAPEs can persist for up to 30 min, fatigue can emerge within 1 min post-exercise [13]. Nonetheless, the research remains inconclusive, with some studies failing to observe improvements in the lower-body performance across a wide range of recovery intervals, from 15 s to 20 min after submaximal back squats [29,30]. Despite this variability, most studies agree that the effective window for an actual performance enhancement is considerably shorter, typically between 3 and 9 min post-conditioning activity [31], though this range may vary substantially due to protocol characteristics and individual recovery profiles [8].

Among the factors influencing these responses, the strength level appears particularly relevant. Although no significant group-by-time interactions were detected in %ΔR, clear group-specific patterns emerged. The SG showed improvements in PAPE10, with %ΔR4 being significantly higher than %ΔR0 (*p* < 0.01, d ≈ 2.5), while the WG responded more favorably in PAPE15, where %ΔR4 exceeded %ΔR0 (*p* < 0.01, d ≈ 2.0), and both groups exhibited marked gains at R8 (*p* < 0.01, d ≈ 2.3–3.1). In the WG, however, these enhancements diminished at R12. These findings align with previous research indicating that stronger individuals tend to achieve peak potentiation later and exhibit a more sustained response compared to their weaker counterparts [12]. The strength level seems to play a key role, as stronger athletes usually recover more rapidly and may show a more robust or prolonged potentiation effect, while weaker athletes are more prone to residual fatigue that can diminish the benefits of the conditioning activity [13,14]. This delay in peak response among stronger athletes may be related to physiological characteristics such as a greater proportion of type II fibers and a higher neuromuscular efficiency [32]. Consequently, stronger athletes, despite experiencing a higher level of fatigue, tend to exhibit greater and longer-lasting potentiation, provided that sufficient recovery is allowed [33]. In line with this, our results suggest that higher-volume protocols were more effective in the WG than in the SG, probably due to the different fatigue-recovery dynamics between groups. Therefore, to optimize the PAPE-induced sprint performance, both the volume of the conditioning stimulus and the recovery duration should be individualized based on the athlete’s strength level and neuromuscular profile [12,13].

The analysis of the CMJ performance revealed that the SG consistently outperformed the WG across all recovery intervals, except at baseline, where no differences were observed. This pattern suggests that strength and neuromuscular differences, initially unapparent, became evident under the influence of the conditioning protocols, reinforcing the notion that stronger individuals are better able to express potentiation, even in the presence of fatigue. The within-group comparisons showed that only the SG significantly improved their CMJ height at R4 relative to R0, while the WG did not exhibit significant enhancements at any recovery point, indicating that potentiation was more evident in and exclusive to stronger athletes. This finding is directly supported by our statistical analysis, where the SG improved significantly at R4 vs. R0 (*p* < 0.05, d ≈ 2.0), whereas the WG showed no significant within-group differences. Prior studies have consistently shown that stronger individuals tend to exhibit greater enhancements in their jump-related performance following heavy-load squats, including greater increases in jump height after a 5RM effort [34], and a superior peak force production in loaded CMJ tests [35]. In contrast, weaker individuals have shown reduced or inconsistent responses when exposed to similar protocols [14]. These findings contrast with the S10 test results, where both the SG and WG exhibited transient improvements, suggesting that the mechanical similarity between the back squat (used as the conditioning activity) and the CMJ may have enhanced the specificity of the PAPE effect [36]. Given that both exercises rely predominantly on vertical force production, potentiation transfer may have been favored in the CMJ, but only when a sufficient force-generating capacity existed, as in the SG. Therefore, beyond absolute strength, mechanical specificity and individual recovery dynamics appear to modulate the effectiveness of PAPEs, highlighting the need for tailored protocols that consider the strength level, fatigue resistance, and biomechanical similarity between the conditioning activity and the performance task [12].

This study is not without limitations. Although median-based grouping has been used in previous research to classify individuals according to their physical fitness level [37], this method may reduce the sensitivity to individual variability in the PAPE response. Moreover, no direct assessments of muscle fiber composition or neuromuscular activation were performed, which limits the understanding of the physiological mechanisms behind performance changes. While classification was based on the bar velocity at a submaximal load, other relevant factors—such as the individual training history, muscle architecture, and fatigue resistance—were not controlled and could have influenced the results. In addition, the CMJ height was estimated using the flight time on an optical system (OptoJump, Microgate, Bolzano, Italy), which, although highly reliable, may over- or underestimate the jump height compared with force plates or a 3D kinematic analysis. Future research should incorporate more detailed physiological assessments, including electromyography and muscle imaging techniques, to better characterize individual differences and their effects on potentiation dynamics.

## 5. Conclusions

The findings of this study suggest that relative strength modulates both the magnitude and time course of PAPE responses in trained high jump athletes. Although no significant three-way interactions were observed among group, PAPE protocol, and recovery interval, both the SG and WG exhibited transient improvements in their S10 and CMJ performance relative to baseline (i.e., percentage change), particularly at 4 (R4) and 8 (R8) min post-activation. However, significant absolute performance improvements were only observed in the SG for their R4 CMJ performance, indicating that the potentiation effect was more robust in stronger athletes. Residual potentiation at 12 min (R12) was found to be both protocol- and group-dependent, with improvements persisting in the SG following PAPE15 and in the WG after PAPE10 relative to the baseline performance. Importantly, the SG consistently outperformed the WG in the CMJ across all time points, highlighting a general advantage in tasks requiring high levels of explosive power. These results emphasize the importance of considering strength levels when implementing PAPE strategies, as they influence both the effectiveness and timing of performance enhancements.

### Practical Applications

Coaches should individualize PAPE protocols based on the athlete’s strength level. In the SG, protocols with lower and higher velocity loss thresholds (e.g., PAPE5 and PAPE15) and medium recovery periods (4–8 min) appear more effective at maximizing performance. In contrast, the WG may benefit from less fatiguing stimuli (e.g., PAPE5 and PAPE10), provided that the recovery is sufficient (≈4 min) to allow potentiation to emerge. However, when more fatiguing protocols such as PAPE15 are employed, a longer recovery period (≥8 min) appears necessary. Furthermore, since the SG showed a consistently superior CMJ performance, developing relative strength should be prioritized, particularly in youth or developmental athletes, to optimize both the acute responsiveness to PAPEs and the long-term explosive capacity.

## Figures and Tables

**Figure 1 jfmk-10-00333-f001:**
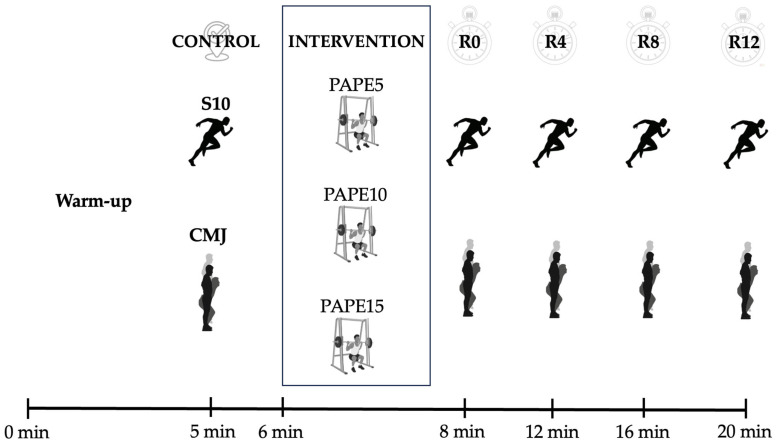
Schematic representation of the experimental design. Note: S10, 10 m sprint test; CMJ, countermovement jump test; PAPE5, intervention protocol based on 5% velocity loss threshold; PAPE10, intervention protocol based on 10% velocity loss threshold; PAPE15, intervention protocol based on 15% velocity loss threshold; R0, measurement taken immediately after the PAPE; R4, measurement taken 4 min after the PAPE; R8, measurement taken 8 min after the PAPE; and R12, measurement taken 12 min after the PAPE.

**Figure 2 jfmk-10-00333-f002:**
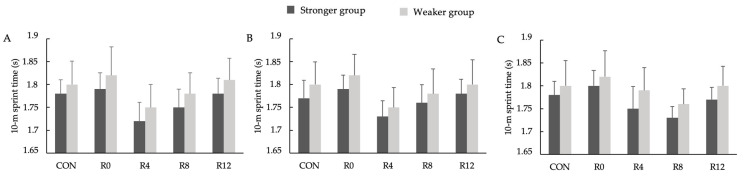
The 10 m sprint time for the stronger group and the weaker group following the control condition without a PAPE (CON) for the 5% velocity loss threshold protocol (**A**), 10% velocity loss threshold protocol (**B**), and 15% velocity loss threshold protocol (**C**).

**Figure 3 jfmk-10-00333-f003:**
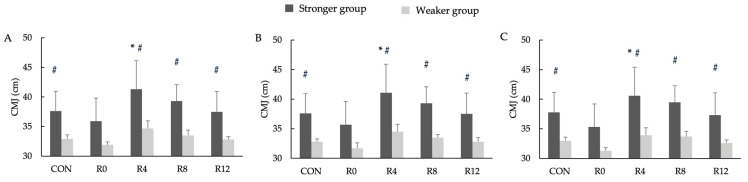
Countermovement jump height for the stronger group and the weaker group following the control condition without a PAPE (CON) for the 5% velocity loss threshold protocol (**A**), 10% velocity loss threshold protocol (**B**), and 15% velocity loss threshold protocol (**C**). Note: * Significant differences with R0 (* *p* < 0.05); # significant differences between stronger group and weaker group (# *p* < 0.05).

**Figure 4 jfmk-10-00333-f004:**
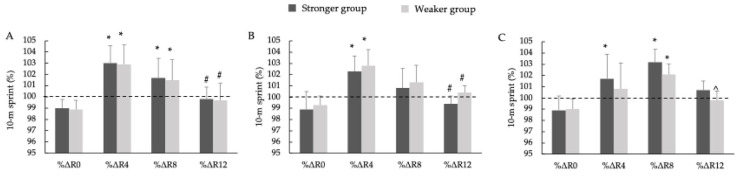
Percentage changes in 10 m sprint time relative to the control condition for the 5% velocity loss threshold protocol (**A**), 10% velocity loss threshold protocol (**B**), and 15% velocity loss threshold protocol (**C**). Note: %Δ represents the percentage change in the sprint time relative to the control condition. The dashed horizontal line marks the baseline (100%) for reference. * Significant differences with %ΔR0 (* *p* < 0.05); # significant differences with %ΔR4 (# *p* < 0.05); ^ significant differences with %ΔR8 (^ *p* < 0.05).

**Figure 5 jfmk-10-00333-f005:**
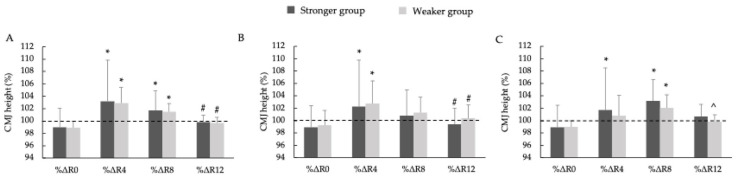
Percentage changes in the countermovement jump height relative to the control condition for the 5% velocity loss threshold protocol (**A**), 10% velocity loss threshold protocol (**B**), and 15% velocity loss threshold protocol (**C**). Note: %Δ represents the percentage change in the sprint time relative to the control condition. The dashed horizontal line marks the baseline (100%) for reference. * Significant differences with %ΔR0 (* *p* < 0.05); # significant differences with %ΔR4 (# *p* < 0.05); ^ significant differences with %ΔR8 (^ *p* < 0.05).

## Data Availability

The data are available from the corresponding authors upon request.

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
