# Peer review of "Strength-Dependent Differences in the Magnitude and Time Course of Post-Activation Performance Enhancement in High Jump Athletes"

_jfmk, 2025, doi:10.3390/jfmk10030333_

Round 1
Reviewer 1 Report
Comments and Suggestions for Authors
I appreciate the opportunity to review this interesting work, which undoubtedly contributes significantly to the field of sports performance. I congratulate the authors for their work.
I have some suggestions that would make the manuscript more robust.
In the introduction, the phrase "Instead, a wide range of complementary factors play a crucial role in an athlete's ability to adapt to and perform under demanding conditions" contains an unnecessary "TO."
In the methodology section, line 209, it would be interesting if you indicated how the maximum heart rate was determined.
Regarding the statistical analysis, it strikes me as odd that you used the Kolmogorov-Smirnov test if the total sample size was 21 participants. The usual approach is to use the Shapiro-Wilk test for samples under 50. If you decide to maintain the Kolmogorov-Smirnov normality test, you should support it with research that supports it.
In the results section, line 279, the phrase "Figure 4 presents the S10 performance results following each intervention." It is confused with the one explained in figure two. I see that one is a percentage and this should be clear in the writing.
Author Response
I appreciate the opportunity to review this interesting work, which undoubtedly contributes significantly to the field of sports performance. I congratulate the authors for their work.
I have some suggestions that would make the manuscript more robust.
AUTHORS RESPONSE (AR): We sincerely thank the reviewer for the positive feedback and for acknowledging the relevance of our work. We also appreciate the constructive suggestions provided, which have been carefully considered. The specific comments have been addressed point by point, and all corresponding changes are highlighted in red in the revised version of the manuscript.
COMMENTS 1: In the introduction, the phrase "Instead, a wide range of complementary factors play a crucial role in an athlete's ability to adapt to and perform under demanding conditions" contains an unnecessary "TO."
AR 1: We appreciate the reviewer’s careful observation. We have removed the unnecessary “to” in the sentence, which now reads: “Instead, a wide range of complementary factors play a crucial role in an athlete’s ability to adapt and perform under demanding conditions” (Line: 39-40)
COMMENTS 2: In the methodology section, line 209, it would be interesting if you indicated how the maximum heart rate was determined.
AR 2: We thank the reviewer for this comment. We have clarified in the methodology section that the maximum heart rate was determined from previous field tests routinely performed by the athletes as part of their physical condition assessment. The revised sentence now reads: “... at an intensity corresponding to 60–70% of the participant’s predicted maximum heart rate, which had been determined from previous field tests (multistage shuttle run test) routinely performed by the athletes as part of their physical condition monitoring” (Line: 212-214)
COMMENTS 3: Regarding the statistical analysis, it strikes me as odd that you used the Kolmogorov-Smirnov test if the total sample size was 21 participants. The usual approach is to use the Shapiro-Wilk test for samples under 50. If you decide to maintain the Kolmogorov-Smirnov normality test, you should support it with research that supports it.
AR 3: We thank the reviewer for pointing this out. Indeed, this was an error in the initial version of the manuscript. As suggested, we have corrected it and now report the use of the Shapiro–Wilk test, which is more appropriate for small sample sizes (n < 50). The revised version of the manuscript reflects this change (highlighted in red).
COMMENTS 4: In the results section, line 279, the phrase "Figure 4 presents the S10 performance results following each intervention." It is confused with the one explained in figure two. I see that one is a percentage and this should be clear in the writing.
AR 4: We thank the reviewer for this observation. We have clarified the description of Figure 4 to avoid confusion with Figure 2. While Figure 2 presents the absolute S10 performance results, Figure 4 presents the percentage changes in S10 performance relative to the control condition. The revised sentence now reads: “Figure 4 presents the percentage changes in S10 performance relative to the control condition following each intervention”
Reviewer 2 Report
Comments and Suggestions for Authors
General comments
This is a really interesting study identifying the benefits of getting strong on PAPE, however, I have some concerns over the categorization of “strong” and actually are the benefits to performance or the PAPE response truly strength characteristics.
Specific comments
Abstract
L14 – A study is inanimate and cannot aim, please amend accordingly.
L14 – As you are not measuring potentiation I would avoid using this term, I would suggest changing to PAPE as already abbreviated.
L17-18 – Is this truly a measure of strength? For some it will be, some it might be endurance. This needs identifying within the limitations.
L25 – What was the magnitude of the transient of relative improvements.
L27 – What do you mean fatigue?
Key words – You have not assessed potentiation; I would remove this. You have identified this is a mechanism that is not for assessing.
Introduction
L44 -What do you mean “neuromuscular activation”
L49 – I am not sure on the power improvements, as they are typically non-performance related improvements.
L58-60 – Probably worth commenting on the fact that these factors have not been clarified as of yet.
L72 – Just as typo aspect, ensure that when reporting r and p they are in italics.
L73 – What was the assessment of relative strength?
L75 – What were the “stronger” players assessed on?
Methods
L102 – Move the reference number to after the name e.g., McKay et al. [17]
L103 – This is a huge limitation of this study, as you may not have actually measured strength. This needs identification in the limitations and also note that you need to report the number of reps lifted, this would aid in identification on the use of this assessment method to determine “strength”
L199-205- Limitation of the use of Optojump and flight time.
Results
Is the enhanced performance purely effect of the group being stronger? If indeed it is strength that is being identified?
Please identify the number of reps performed and what the loads were that were lifted between each group?
Discussion
L322 - A study is inanimate and cannot aim, please amend accordingly.
L323 – Change 10-m sprint to S10 as already abbreviated.
The discussion of the study is well written and developed with key considerations and implications, however, more needs to be identified within the results and discussion around the measurements made and results presented.
Author Response
This is a really interesting study identifying the benefits of getting strong on PAPE, however, I have some concerns over the categorization of “strong” and actually are the benefits to performance or the PAPE response truly strength characteristics.
Authors Response (AR): We sincerely thank the reviewer for the positive feedback and for raising this important point regarding the categorization of "strong" athletes and whether the benefits observed reflect true strength-related characteristics. We agree that this is a key aspect to clarify. Accordingly, we have revised the manuscript to better justify our criteria for dividing participants into stronger and weaker groups, and we have expanded the discussion to highlight how relative strength may influence the PAPE response rather than absolute performance per se. All changes addressing this comment can be found in the revised version of the manuscript, marked in red for ease of identification.
Specific comments
Abstract
COMMENTS 1: L14 – A study is inanimate and cannot aim, please amend accordingly.
AR 1: We thank the reviewer for this observation. Following the suggestion, we have amended the sentence in the abstract to avoid attributing intention to the study. The sentence now reads: “The aim of this study was to analyze the potentiation timeline following three PAPE protocols, considering strength level as a modulating factor in trained high jump athletes” (Line 14-16).
COMMENTS 2: L14 – As you are not measuring potentiation I would avoid using this term, I would suggest changing to PAPE as already abbreviated.
AR 2: We thank the reviewer for this comment. Building on the revision made in Comment 6, we have further refined the abstract to avoid the term "potentiation," which does not directly reflect our measurements. It has been replaced with "performance responses," resulting in the following updated sentence:
"The aim of this study was to analyze the performance responses following three PAPE protocols, considering strength level as a modulating factor in trained high jump athletes"
COMMENTS 3: L17-18 – Is this truly a measure of strength? For some it will be, some it might be endurance. This needs identifying within the limitations.
AR 3: We thank the reviewer for this comment. The division of athletes into Stronger and Weaker groups was based on the load lifted at 0.8 m/s in a velocity-based half-squat test. This test involves lifting a relatively heavy load at a controlled high velocity with very few repetitions, which specifically targets strength and power capabilities. Due to the low number of repetitions and the nature of the load-velocity relationship, the test does not impose a significant endurance demand and therefore does not reflect muscular endurance. In this context, the median load at 0.8 m/s provides a valid and widely used indicator to categorize athletes according to their strength/power characteristics [Sanchez-Sanchez J, Rodriguez A, Petisco C, et al. Effects of Different Post-Activation Potentiation Warm-Ups on Repeated Sprint Ability in Soccer Players from Different Competitive Levels. J Hum Kinet 2018; 61: 189–197]
COMMENTS 4: L25 – What was the magnitude of the transient of relative improvements.
AR 4: We thank the reviewer for this valuable comment. In the revised abstract, we have clarified both the statistical significance and the magnitude of the transient improvements, in line with the results reported in the manuscript. The sentence now reads: “However, both groups exhibited transient improvements in S10 and CMJ performance that were statistically significant (p < 0.05) and of large magnitude (d = 1.93–3.15), observed at 4 and/or 8 min post-activation, which subsequently declined by 12 min” (Line 24-27).
COMMENTS 5: L27 – What do you mean fatigue?
RESPONSE 5: We thank the reviewer for this comment. To avoid ambiguity associated with the term “fatigue,” we have replaced it with “less and more demanding” to describe the protocols. This terminology more accurately reflects the degree of stimulus induced by the velocity loss thresholds (5% to 15%) and maintains clarity regarding the interpretation of the results. The sentence now reads: “Strength level modulates both the time course and magnitude of PAPE. Stronger athletes responded better to both less and more demanding protocols (5% to 15% velocity loss thresholds) with 4–8 min recovery, whereas weaker athletes benefited mainly from less demanding stimuli (5% velocity loss thresholds), provided that recovery is sufficient (≈4 min) to allow potentiation to emerge. However, with more demanding protocols (15% velocity loss thresholds), a longer recovery period (≈8 min) appears necessary”
COMMENTS 6: Key words – You have not assessed potentiation; I would remove this. You have identified this is a mechanism that is not for assessing.
AR 6: We thank the reviewer for this observation. To accurately reflect the scope of our study, we have removed “Post-Activation Potentiation” from the keywords. Instead, we have included “Acute Performance Response” to capture the transient changes in sprint and jump performance measured following the PAPE protocols.
Introduction
COMMENTS 7: L44 -What do you mean “neuromuscular activation”
AR 7: We thank the reviewer for this comment. To clarify the meaning of “neuromuscular activation” we have revised the sentence to specify that these are brief pre-activation protocols designed to acutely enhance neuromuscular function, thereby preparing the muscles and nervous system for explosive and high-intensity actions. This change provides a clearer description of the strategies assessed in our study
COMMENTS 8: L49 – I am not sure on the power improvements, as they are typically non-performance related improvements.
AR 8: We thank the reviewer for this comment. To clarify, we have revised the sentence to refer more generally to “performance outcomes” instead of specifying particular actions. This reflects the general nature of the cited literature and accurately describes the potential effects of PAPE protocols without implying measurement of variables not assessed in our study
COMMENTS 9: L58-60 – Probably worth commenting on the fact that these factors have not been clarified as of yet.
AR 9: We thank the reviewer for this comment. We have revised the sentence to acknowledge that the physiological mechanisms underlying this effect are not yet fully understood. The phrase now reads: “This effect is believed to result from several physiological mechanisms, which are not yet fully understood, including the phosphorylation of myosin regulatory light chains—which may increase calcium sensitivity and enhance force production” (Line58-60)
COMMENTS 10: L72 – Just as typo aspect, ensure that when reporting r and p they are in italics.
AR 10: We thank the reviewer for pointing this out. In the revised manuscript, all statistical symbols, including r and p, have been formatted in italics in accordance with journal style guidelines
COMMENTS 11: L73 – What was the assessment of relative strength?
AR 11: We thank the reviewer for this comment. To clarify, in the study by Jo et al., relative strength was calculated as the one-repetition maximum (1RM) in the back squat divided by body mass. We have updated the sentence in the Introduction to specify this method.
COMMENTS 12: L75 – What were the “stronger” players assessed on?
AR 12: We thank the reviewer for this comment. We have clarified in the manuscript that in the cited study, rugby players were divided into stronger and weaker groups based on relative one-repetition maximum (1RM) in the back squat. Strong players had a relative 1RM ≥ 2 × body mass, whereas weak players had a relative 1RM < 2 × body mass. This clarification specifies how strength was assessed and defines the groups, providing context for the reported differences in potentiation timing and magnitude
Methods
COMMENTS 13: L102 – Move the reference number to after the name e.g., McKay et al. [17]
AR 13: We thank the reviewer for this comment. All reference numbers have been moved to follow the author names in accordance with the journal’s citation style, for example: McKay et al. [17]."
COMMENTS 14: L103 – This is a huge limitation of this study, as you may not have actually measured strength. This needs identification in the limitations and also note that you need to report the number of reps lifted, this would aid in identification on the use of this assessment method to determine “strength”
AR 14: We appreciate the reviewer’s observation. However, we respectfully disagree with the comment that strength was not properly measured in our study. Previous research (González-Badillo et al., 2010) has established that mean propulsive velocity (MPV) is a valid method to estimate 1RM and the corresponding %1RM associated with different loads. Accordingly, in our methodology we used a progressive half-squat test on a Smith machine to determine the individualized load associated with a bar velocity of 0.8 m/s. This testing protocol has been validated and widely used in the literature as an indirect but reliable indicator of maximal strength levels.
In our procedure, participants performed sets of 3 continuous repetitions per load increment, with the test concluding when the athlete was no longer able to reach an MPV ≥ 0.8 m/s across all three repetitions. This approach allows accurate identification of the load associated with the target velocity while minimizing the influence of fatigue. Therefore, reporting the total number of repetitions performed across the progressive test would not provide meaningful additional information regarding the strength assessment, since the criterion variable was the bar velocity achieved with a given load, not the absolute number of repetitions.
Based on these considerations, we believe that the method employed in our study is an appropriate and scientifically supported approach to classify participants into stronger and weaker groups, and does provide a valid indicator of individual strength capacity.
González-Badillo JJ, Sánchez-Medina L. Movement velocity as a measure of loading intensity in resistance training. Int J Sports Med 2010; 31: 347–352.
COMMENTS 15: L199-205- Limitation of the use of Optojump and flight time.
AR 15: We acknowledge the reviewer’s observation. We thank the reviewer for this valuable observation. We acknowledge that using an optical system such as OptoJump, which estimates jump height from flight time, may present limitations compared with force platforms or 3D motion capture. To address this, we have now included this point in the limitations section of the manuscript: “In addition, CMJ height was estimated using flight time on an optical system (OptoJump), which, although highly reliable, may over- or underestimate jump height compared with force plates or 3D kinematic analysis” (Line 422-425).
Results
COMMENTS 16: Is the enhanced performance purely effect of the group being stronger? If indeed it is strength that is being identified?
AR 16: We appreciate this important observation. Strength is indeed considered a key determinant of the magnitude and time course of PAPE. Previous studies (Jo et al., 2010) have consistently reported that stronger athletes, often defined relative to their body mass, show greater potentiation and earlier performance peaks compared to weaker athletes under the same conditioning protocols. This suggests that maximal strength is not only a descriptor of performance capacity, but also a conditioning variable that modulates the responsiveness to activation strategies. In this context, the differences observed between our groups can be reasonably interpreted as being influenced by their strength levels, which is in line with prior evidence indicating that strength status is a conditioning factor in the efficacy of PAPE responses.
Jo E, Judelson AA, Brown EE, et al. Influence of recovery duration after a potentiating stimulus on muscular power in recreationally trained individuals. J Strength Cond Res 2010; 2: 343–347.
COMMENTS 17: Please identify the number of reps performed and what the loads were that were lifted between each group?
AR 17: We thank the reviewer for this comment. The load lifted at 0.8 m/s, which was used to individualize the conditioning stimulus, is already reported in the Participants section and was the criterion to classify athletes into the Stronger and Weaker groups. In addition, we now provide the number of repetitions performed by each group in the different PAPE protocols. Specifically, in PAPE5 the Stronger Group completed 5.70 ± 1.42 repetitions while the Weaker Group completed 6.18 ± 1.83; in PAPE10 the Stronger Group performed 9.10 ± 1.97 repetitions versus 10.3 ± 2.33 in the Weaker Group; and in PAPE15 the Stronger Group achieved 12.2 ± 2.30 repetitions compared to 12.6 ± 2.77 in the Weaker Group.
Discussion
COMMENTS 18: L322 - A study is inanimate and cannot aim, please amend accordingly.
AR 18: We thank the reviewer for this observation. The wording has been revised to avoid attributing intentionality to the study.
COMMENTS 19: L323 – Change 10-m sprint to S10 as already abbreviated.
AR 19: This point was resolved with the new formulation of the study objective, so no further modification was required.
COMMENTS 20: The discussion of the study is well written and developed with key considerations and implications, however, more needs to be identified within the results and discussion around the measurements made and results presented.
AR 20: We thank the reviewer for this valuable observation. In the revised version, we have expanded the Discussion to more explicitly connect the statistical results with their interpretation. Specifically, we now include direct references to the significant within- and between-group comparisons, effect sizes, and the recovery-time patterns observed in both S10 and CMJ tests (e.g., significant improvements at R4 and R8 in PAPE5 for both groups; significant gains at R4 in SG and WG during PAPE10; exclusive improvements for WG at R4 and R8 in PAPE15; significant within-group enhancement in CMJ at R4 only in SG). These details were added to clarify how the measurements and statistical outcomes support the practical and theoretical implications discussed. We believe this addition strengthens the alignment between the Results and Discussion sections, addressing the reviewer’s concern.
Round 2
Reviewer 1 Report
Comments and Suggestions for Authors
I congratulate the authors for the excellent work done on this research. I consider the topic to be extremely relevant and also allows for the generation of future lines of research. I have no suggestions for changes in this round.